High expression of HSP90 is associated with poor prognosis in patients with colorectal cancer

Zhang Shuming 1
Guo Shichao 1
Li Zhangfu 1
Li Dan 1 eileenld@gmail.com
Zhan Qimin 1 2 zhanqimin@bjmu.edu.cn
1 State Key Laboratory of Molecular Oncology, National Cancer Center/National Clinical Research Center for Cancer/Cancer Hospital, Chinese Academy of Medical Sciences and Peking Union Medical College , Beijing , China
2 Laboratory of Molecular Oncology, Key laboratory of Carcinogenesis and Translational Research (Ministry of Education/Beijing), Laboratory of Molecular Oncology, Peking University Cancer Hospital & Institute , Beijing , China
Zhao Min
Electronic publication date: 2019 Oct 31
Publication date: 2019
Volume: 7
Electronic Location ID: e7946
Received 2019 Jun 23; Accepted 2019 Sep 24
Copyright: © 2019 Zhang et al.
Copyright year: 2019
Copyright holder: Zhang et al.
License: This is an open access article distributed under the terms of the Creative Commons Attribution License, which permits unrestricted use, distribution, reproduction and adaptation in any medium and for any purpose provided that it is properly attributed. For attribution, the original author(s), title, publication source (PeerJ) and either DOI or URL of the article must be cited.
License URL: https://creativecommons.org/licenses/by/4.0/

Keywords: HSP90, Colorectal cancer, Outcomes, Prognosis, Immunohistochemistry, Proteasome, Base excision repair, Overall survival

Funding: National 973 Program 2015CB553904 National Natural Science Foundation of China 81490753, 81830086 and 81672455 Beijing Nova Program xx2018040 CAMS Initiative for Innovative Medicine 2017-I2M-3-004 This work was supported by the National 973 Program (2015CB553904), the National Natural Science Foundation of China (81490753, 81830086 and 81672455), the Beijing Nova Program (xx2018040) and the CAMS Initiative for Innovative Medicine (2017-I2M-3-004). The funders had no role in study design, data collection and analysis, decision to publish, or preparation of the manuscript.

==============================
Background

Heat shock protein 90 (HSP90) is a highly conserved chaperone with an approximate molecular weight of 90-kDa. It plays a critical role in maintaining stability and homeostasis of oncoproteins, helping cancer cells living in the unsuitable environmental conditions. The current study aims to inquire the difference of HSP90 expression in tumor tissues and normal tissues, analyze the correlation between HSP90 expression and the prognoses of patients with colorectal cancer (CRC), and investigate its role in CRC preliminarily.

Methods

Online analysis of HSP90 mRNA levels in different cancers was firstly done in Gene Expression Profiling Interactive Analysis. Then HSP90 expression was determined by immunohistochemistry between 99 CRC tissues and 81 normal tissues. Chi-square test or Fisher’s exact test was used to analyze the relationship between HSP90 and histopathologic characteristics. Kaplan–Meier analysis and Cox’s proportional hazards model were also done for further analysis of the prognostic values of HSP90. Pearson’s correlation coefficients between HSP90 expression values and other mRNA expression values were calculated based on The Cancer Genome Atlas dataset and bioinformatic analysis was done about these screened genes.

Results

Colorectal cancer tissues showed significantly higher expression of HSP90 than normal tissues (55.6% vs. 3.7%, P < 0.0001). Kaplan–Meier curves showed high HSP90 expression was associated with poor prognosis (P = 0.039) in CRC patients, and multivariate Cox proportional hazards regression model analysis also indicated that HSP90 expression (HR = 1.930, 95% CI [1.113–3.349], P = 0.019) linked to poor prognosis. Moreover, 85 genes were correlated with HSP90, which were involved in metabolic process and enriched in pathways of Proteasome and Base excision repair.

Conclusions

Our results suggested that HSP90 expression is inversely associated with survival outcomes and could be an independent prognostic factor for CRC patients. It mainly involved in metabolic process and exerted binding and catalytic activities.

Introduction

Colorectal cancer (CRC) is the third leading malignancy for both genders combined, accounting for approximately 9.2% cancer-related deaths worldwide. Epidemiologic studies showed an increasing trend in the overall CRC incidence and mortality with an estimated 1,849,518 new cases in 2018 (Ferlay et al., 2019). Surgical resection is regarded as the standard treatment for locally CRC, while adjuvant therapy and neoadjuvant therapy are strongly suggested for patients with distant metastasis tumors (Recio-Boiles & Cagir, 2019). The prevalence of screening programs for the early detection of CRC patients and novel therapeutic strategies in metastatic patients have greatly improved the general outcomes (Siegel et al., 2017). However, approximately half of the patients will die within 5 years after diagnosis, especially for patients diagnosed at stage IV, which have a 5-year overall survival rate at around only 10% (Kuipers et al., 2015). Therefore, it is important to identify the prognostic biomarkers for the optimized treatments and development of new therapeutic strategies.

Heat shock protein 90 (HSP90) is one of the most abundant proteins in normal cells, which plays central roles in many signaling and other cellular pathways (Makhnevych & Houry, 2012). Previous studies have shown that its expression is significantly higher in tumors than normal cells, helping cancer cells to survive in poor physiological and stress conditions such as hypoxia, nutritional deficiencies, low pH, and exposure to UV light and chemicals (Whitesell & Lindquist, 2005). Along with other chaperones, HSP90 participated in the stabilization and maturation of a large number of client proteins, ranging from stress regulation and protein folding to DNA repair, development, the immune response and many other processes (Echeverría et al., 2011). Moreover, many of the oncoproteins that contribute to accelerated growth, proliferation and survival of neoplastic cells are client proteins of the HSP90 chaperone complex, including Bcr-Abl, HER-2, EGFR, C-Raf, B-Raf, Akt, Met, VEGFR, FLT3, AR, and HIF-1α (Morán et al., 2019). Thus, HSP90 is an attractive target for drug design and there are already many clinical trials studying the role of HSP90 inhibitors in the process of cancer therapies (Jhaveri et al., 2014).

In this present study, we aimed to demonstrate the correlation between HSP90 and outcomes of CRC. To achieve this goal, we firstly analyzed the expression of HSP90 protein based on the Gene Expression Profiling Interactive Analysis (GEPIA) database and the results showed a higher expression of HSP90 in various types of cancers including CRC. In addition, immunohistochemistry analysis of paraffin-embedded specimens from 99 patients also proved the up-regulation of HSP90 protein in CRC tissues comparing with adjacent tissue. We further investigated its associations with clinicopathologic factors and prognosis, and the results showed the increased HSP90 expression was related to poor prognosis of CRC patients. Moreover, the main biological pathways that related with HSP90 expression in CRC were analyzed by using Gene Ontology (GO) and the Kyoto Encyclopedia of Genes and Genomes (KEGG) analysis.

Materials and Methods

Patients

Formalin-fixed paraffin-embedded tumor specimens from surgical resections of the primary tumor were obtained from 99 patients with CRC who underwent curative surgery at our hospital from September 2008 to August 2010. The ethical approval of the study was obtained from the ethics committee of National Cancer Center/Cancer Hospital of Chinese Academy of Medical Sciences and Peking Union Medical College (NCC2018-071). These patients had confirmed pathological diagnosis and complete clinical data, and did not receive radiotherapy or chemotherapy before the surgery. Survival data of all these patients were followed up by telephone interview until October 31, 2018.

Immunohistochemistry and evaluation of staining

Immunohistochemistry assay was performed with four µm sections of paraffin-embedded tissues on polylysine coated slides. Slides were deparaffinized with xylene and rehydrated through descending gradient ethanol (100–95–85–75%). Then these slides were washed with phosphate-buffered saline solution (PBS, 0.01M, pH 7.0) for three times. Antigen retrieval was performed by boiling in 10 mM sodium citrate buffer (pH 6.0) for 15 mins and then cooled down to room temperature (RT) naturally. Endogenous peroxidase activity was blocked with 3% hydrogen peroxide at RT for 10 mins. After being washed for three times with PBS, the sections were incubated in 5% bovine serum albumin at RT for 30 min to block nonspecific binding. Then they were incubated with anti-HSP90 mAb (1:500) (cat: ab59459; Abcam, Cambridge, MA, USA) overnight at 4 °C. After washing with PBS for three times, the specimens were incubated with horseradish peroxidase-conjugated goat anti-rabbit IgG secondary antibodies (1:500) (cat: no. WLA023a; Wanleibio Co., Ltd., Shenyang, China) for 1 h at RT and visualized by DAB development with Dako EnVison kit (Dako, Glostrup, Denmark). Finally, all slides were counterstained with hematoxylin, dehydrated and mounted with glycerol gelatin. Sections were observed with a microscope.

The HSP90 expression-intensity scores were independently determined by two experienced pathologists who were blinded to the information of patients. According to the percentage of the HSP90 positive tumor cells and intensity of staining, the expression area was graded into four levels: 0 (0), 1 (<1/10), 2 (1/10–1/3), 3 (1/3–2/3), and 4 (≥2/3). The staining intensity score was defined as follows: no staining, 0; weak staining, 1; moderate staining, 2; strong staining, 3. The total HSP90 expression index was generated by multiplying the score of staining intensity and the percentage of HSP90 expression, which ranged from 0 to 12. We defined scores of 0–3 as the low expression group, and scores of 4–12 as the high expression group.

Analysis of HSP90 genes expression level in different cancers

Online analysis of HSP90 mRNA levels of different cancers was done by GEPIA, http://gepia.cancer-pku.cn/index.html), a web server for cancer and normal gene expression profiling analysis. It worked based on database of the The Cancer Genome Atlas (TCGA) and the GTEx project including 9,736 tumors and 8,587 normal samples RNA-Seq data and could be used to analyze gene expression through a standard processing pipeline.

Statistical analysis

Nonparametric Mann–Whitney U test was used to detect the HSP90 expression difference between cancer tissues and adjacent normal tissues. Chi-square test or Fisher’s exact test was performed to analyze the relationship between HSP90 status and CRC patients’ clinicopathological features. Survival curves were determined with the Kaplan–Meier method, and different survival rates between groups were compared with the log-rank test. The significance of variables for survival was conducted with the Cox proportional hazards model in univariate and multivariate analysis. All statistical analysis was performed with the SPSS 13.0 statistical software (IBM Corp., Armonk, NY, USA). P < 0.05 (two-tailed) was considered to indicate a statistically significant difference.

Bioinformatics analyses

We firstly downloaded RNA Seq V2 RSEM data which including 17,989 gene expression of 382 CRC tissues from https://www.cbioportal.org/ by using R “cgdsr” package. By calculating Pearson’s correlation coefficients, we screened genes which positively correlated with the expression of HSP90 (Pearson’s correlation ≥ 0. 4, P < 0.0001). Then GO analysis and KEGG analysis were performed using edgeR on OmicShare, an online platform for data analysis (www.omicshare.com/tools). Q value < 0.05 was used as the thresholds in selecting significant GO and KEGG pathways.

Results

Dataset analysis indicated a significant different expression of HSP90 between cancer tissues and normal tissues

Gene Expression Profiling Interactive Analysis is a commodious and intuitive online tool for gene analysis, a web-based tool based on TCGA and GTEx data. It provides key interactive and customizable functions including differential genes expression analysis, profiling plotting, similar gene detection, correlation analysis, and dimensionality reduction analysis (Tang et al., 2017). We analyzed the HSP90 mRNA expression profile across all tumor samples and normal tissues in GEPIA. From these results showed in bar plot (Fig. 1A), we found HSP90 had a different expression between some kinds of cancers and normal tissues. Statistical analysis showed in dot plot indicated significant higher expression in 10 kinds of cancer tissues: breast invasive carcinoma, cervical squamous cell carcinoma and endocervical adenocarcinoma, colon adenocarcinoma (COAD), lymphoid neoplasm diffuse large B-cell lymphoma, esophageal carcinoma, pancreatic adenocarcinoma, rectum adenocarcinoma (READ), skin cutaneous melanoma (SKCM), stomach adenocarcinoma, and thymoma (THYM) than corresponding normal tissues (Fig. 1B). Detailed analyses of the expression of HSP90 of COAD and READ with TCGA data were shown with number of samples in Fig. 1C.

Figure 1 Analysis of HSP90 genes expression level s from online database.

HSP90 gene expression profile (mRNA level) across all tumor samples and normal tissues were analyzed in GEPIA and these results were shown in bar plot (A) and dot plot (B) separately. (C) Expression levels of HSP90 gene in rectal cancers (READ), colon cancers (COAD) and normal tissues were analyzed using TCGA data. The statistical analysis was performed by using the Student's t test. *P < 0.05. Abbreviations: ACC, Adrenocortical carcinoma; BLCA, bladder urothelial carcinoma; BRCA, breast invasive carcinoma; CESC, cervical squamous cell carcinoma and endocervical adenocarcinoma; CHOL, cholangio carcinoma; COAD, colon adenocarcinoma; DLBC, lymphoid neoplasm diffuse large B-cell lymphoma; ESCA, esophageal carcinoma; GBM, glioblastoma multiforme; HNSC, head and neck squamous cell carcinoma; KICH, kidney chromophobe; KIRC, kidney renal clear cell carcinoma; KIRP, kidney renal papillary cell carcinoma; LAML, acute myeloid leukemia; LGG, brain lower grade glioma; LIHC, liver hepatocellular carcinoma; LUAD, lung adenocarcinoma; LUSC, lung squamous cell carcinoma; OV, ovarian serous cystadenocarcinoma; PAAD, pancreatic adenocarcinoma; PCPG, pheochromocytoma and paraganglioma; PRAD, prostate adenocarcinoma; SARC, sarcoma; SKCM, skin cutaneous melanoma; STAD, stomach adenocarcinoma; TGCT, testicular germ cell tumors; THCA, thyroid carcinoma; THYM, thymoma;) UCEC, uterine corpus endometrial carcinoma; UCS, uterine carcinosarcoma; READ, rectum adenocarcinoma.

HSP90 had a higher expression in colon cancer tissues

Immunohistochemistry was done to evaluate and compare the expression of HSP90 between 99 CRC tissues and 81 adjacent tissues. The expression of HSP90 mainly located in cytoplasm, which had no significant difference between cancer tissues and adjacent tissues (Fig. 2). The expressions of HSP90, evaluated by the HSP90 expression score index, were less than four points in most normal tissues and were lower than cancer tissues.

Figure 2 Immunohistochemistry and statistical results of HSP90 expression.

(A–H) The expression of HSP90 in colonic cancer and matched adjacent normal epithelial tissues was detected by immunohistochemistry. Original magnification: 40× and 200×. (I) Statistical analysis was done by comparing the final scores in colon cancer and adjacent normal epithelial tissues. Colon cancer tissues had a significant higher expression than matched adjacent normal epithelial tissues. The statistical analysis was performed by using the Student’s t-test. ***P < 0.001.

In order to make a deeper analysis, we defined score index 0 to 3 as low-level group (including negative expression), 4 to 12 as high-level group. In 99 CRC tissues, 55 samples (55.6%) were in the high-level group and the rest (44.4%) were in the low-level group. In contrary, 78 out of 81 (96.3%) adjacent tissues were in the low-level group. Furthermore, 65 out of 78 (80.2%) adjacent tissues were negative expression. Statistical analysis showed a significantly higher expression of HSP90 in cancer tissues (Nonparametric Mann–Whitney U test, P < 0.0001) (Fig. 2B).

Association between HSP90 expression and clinicopathological characteristics of CRC patients

Then we collected clinicopathological parameters of CRC patients and investigated correlations with HSP90 expression (Table 1). As for general clinical characteristics, no significant expression difference was found in sex groups (P = 0.223) or age groups (P = 0.438). Chi-square test and Fisher’s exact test were also used to analyze the relationship between HSP90 and some pathological variables which were important for evaluating prognosis. However, no obvious association was found between HSP90 expression levels in CRC tissues and tumor location (P = 0.604), pathological differentiation (P = 0.621), T phase (P = 0.505), lymph node metastasis (P = 0.608) or American Joint Committee on Cancer (AJCC) stage (P = 0. 749).

Table 1 Association between HSP90 expression and clinicopathological characteristics of patients.

Characteristics	Total cases (%)	HSP90 expression	P-value	
Low (%)	High (%)	
Gender				0.223	
Male	54 (54.5%)	27 (50.0%)	27 (50.0%)		
Female	45 (45.5%)	17 (37.8%)	28 (62.2%)		
Age (year)				0.438	
<65	31 (31.3%)	12 (38.7%)	19 (61.3%)		
≥65	68 (68.7%)	32 (47.1%)	36 (52.9%)		
Location				0.604	
Ascending colon	35 (35.4%)	17 (48.6%)	18 (51.4%)		
Transverse colon	25 (25.3%)	9 (36.0%)	16 (64.0%)		
Descending-Sigmoid colon	39 (39.4%)	18 (46.2%)	21 (53.8%)		
T category				0.505	
T1 + T2	10 (10.1%)	3 (30.0%)	7 (70.0%)		
T3 + T4	89 (89.9%)	41 (46.1%)	48 (53.9%)		
N category				0.608	
N0	59 (59.6%)	27 (45.8%)	32 (54.2%)		
N1	29 (29.3%)	11 (37.9%)	18 (62.1%)		
N2	11 (11.1%)	6 (54.5%)	5 (45.5%)		
Differentiation				0.621	
Well to moderate	49 (49.5%)	23 (46.9%)	26 (53.1%)		
Low	50 (50.5%)	21 (42.0%)	29 (58.0%)		
AJCC stage				0.749	
I, II	59 (59.6%)	27 (45.8%)	32 (54.2%)		
III, IV	40 (40.4%)	17 (42.5%)	23 (57.5%)		
Overall survival				0.039	
Death	59 (59.6%)	21 (35.6%)	38 (64.4%)		
Alive	40 (40.4%)	23 (57.5%)	17 (42.5%)		

High expression of HSP90 predicts poor prognosis of CRC patients

Survival analysis was conducted to explore whether HSP90 expression influences clinical progression in CRC patients. Here, we did not analyze the relationship between M category (describes the presence or otherwise of distant metastatic spread) and prognosis as only three cases had distal metastasis. Besides lymph node metastasis (P < 0.001), AJCC stage (P < 0.001) and pathological differentiation (P = 0.033), high HSP90 expression was associated with poor prognosis (P = 0.039) in CRC patients as well. HSP90 high-expression group have a significantly worse overall mean of survival (OS) than low-expression group (51.6 vs. 66.0 months) (Fig. 3).

Figure 3 Kaplan–Meier survival analysis in subsets of CRC patients.

(A) Kaplan–Meier survival analysis of CRC patients according to lymph node metastasis. (B) Kaplan–Meier survival analysis of CRC patients according to AJCC stage. (C) Kaplan–Meier survival analysis of CRC patients according to pathological differentiation grade. (D) Kaplan–Meier survival analysis of CRC patients according to HSP90 expression.

Next, Cox’s proportional hazards model was also done for further analysis of prognostic value of HSP90 expression and other clinical parameters (Table 2). These results revealed that variables including lymph node metastasis (HR = 2.192, 95% CI [1.544–3.112], P < 0.001), pathological differentiation (HR = 1.737, 95% CI [1.035–2.915], P = 0.037) and high HSP90 expression (HR = 1.738, 95% CI [1.018–2.966], P = 0.043) linked to poor prognosis (Fig. 4A). Then, multivariate analysis of Cox’s proportional hazards regression model was further done to exclude the false positive results caused by the interaction of different factors. Multivariate analysis results revealed that, in addition to the acknowledgeable factors including lymph node metastasis (HR = 2.459, 95% CI [1.652–3.661], P < 0.001) and pathological differentiation (HR = 1.749, 95% CI [1.017–3.007], P = 0.043), HSP90 expression level (HR = 1.930, 95% CI [1.113–3.349], P = 0.019) could be an independent prognostic factor for CRC patient prognosis prediction (Fig. 4B).

Figure 4 Forest graphs of Cox proportional hazards model analysis of variables affecting overall survival in colorectal cancer patients.

(A) Univariate analysis results; (B) Multivariate analysis results.

Table 2 Cox proportional hazards model analysis of variables affecting overall survival in colorectal cancer patients.

Variable	Categories	Univariate analysis	Multivariate analysis	
HR (95% CI)	P	HR (95% CI)	P	
Gender	Male vs. Female	1.004 [0.600–1.679]	0.988	0.620 [0.344–1.119]	0.113	
Age (years)	<65 vs. ≥65	1.542 [0.858–2.773]	0.148	1.756 [0.957–3.221]	0.069	
Differentiation	Poor vs. Well + moderate	1.737 [1.035–2.915]	0.037	1.749 [1.017–3.007]	0.043	
T category	T1 + T2 vs. T3 + T4	1.288 [0.515–3.221]	0.589	1.128 [0.416–3.058]	0.813	
N category	N0 vs. N1 vs. N2	2.192 [1.544–3.112]	0.000	2.459 [1.652–3.661]	0.000	
Location	Ascending vs. Transverse vs. Descending-Sigmoid	0.773 [0.576–1.037]	0.086	0.899 [0.658–1.227]	0.501	
HSP90	High vs. Low	1.738 [1.018–2.966]	0.043	1.930 [1.113–3.349]	0.019	
Note:

HR, hazard ratio; 95% CI, 95% confidence interval.

HSP90-related signaling pathways based on TCGA database

We performed functional enrichment analysis to explore the biological pathway and process correlated with HSP90. Firstly, the Pearson’s correlation coefficients between HSP90 expression values and other mRNA expression values were calculated in the TCGA dataset. A total of 85 genes were positively correlated with the expression of HSP90 (defined as Pearson’s correlation coefficient ≥ 0. 4, as shown in Data S1) and included in the GO analysis (Fig. 5) and the KEGG analysis, respectively (as shown in Supplemental Data). The GO analysis results showed that it mainly existed as an intracellular molecular (Fig. 5D), which were involved in metabolic process (Fig. 5B) and exerted binding and catalytic activity (Fig. 5C). Besides, KEGG analysis enriched these genes in pathways of Proteasome and Base excision repair (Fig. 6).

Figure 5 Enriched GO analysis of identified genes.

(A) The results are summarized in three main categories: biological processes, molecular functions and cellular components. (B) The top 20 enriched GO biological processes terms. (C) The top 20 enriched GO molecular functions terms. (D) The top 20 enriched GO cellular components terms.

Figure 6 KEGG pathway classification of identified genes.

(A) Distribution of the KEGG pathways of identified genes is shown as a bar chart. The horizontal axis is the number of genes, whereas the vertical ordinates are the terms of the KEGG pathways. Q < 0.05 was used as the thresholds in selecting significant KEGG pathways. (B) The significant pathways of KEGG enrichment.

Discussion

Colorectal cancer was one of the most frequent tumor types worldwide and approximately half of the patients will die within 5 years after diagnosis (Siegel, Miller & Jemal, 2018). Therefore, it is of great importance to identify novel biomarkers and therapeutic targets, for the improvement of clinical outcomes for CRC patients. Hsp90 is a highly conserved and ubiquitously expressed molecular chaperone that involved in the folding, maturation, and degradation of client proteins (Sima & Richter, 2018). It participates in essential cellular activities by supporting the maturation process of its client proteins, most of which are involved in cell growth, proliferation and survival, which are critical functions for neoplastic cells (Kryeziu et al., 2019).

In this study, the HSP90 mRNA expression across all tumor samples and normal tissues based on GEPIA dataset were analyzed. The results indicated significant elevated expressions in multiple cancer types including CRC, consistent with the previous studies that reported elevated HSP90 expressions in breast cancer, lung cancer, melanoma and myeloid leukemia (Yano et al., 1999; Becker et al., 2004; Biaoxue et al., 2012; Gonzalez et al., 2014). Additionally, a higher protein expression level of HSP90 was further observed in 99 CRC tissues compared with 81 adjacent normal tissues by IHC assay. The increased HSP90 expression was significantly associated with poor prognosis of CRC patients, revealed by survival curve analysis. Our results indicated both mRNA and protein levels of HSP90 were elevated in tumor tissues, which suggested that HSP90 gene had a higher activity in both transcriptional and translational levels in CRC tissues.

To further investigate the association between the clinical characteristics and prognosis, Cox’s regression analysis was performed. The results suggested that tumor lymph node metastasis, differentiation status, AJCC stage (American Joint Committee on Cancer, an organization best known for defining and popularizing cancer staging standards according, officially the AJCC staging system) and HSP90 expression were the independent risk factors impacting the OS of CRC patients. Therefore, we speculated that HSP90 might act as an oncogene and could be considered as a potential therapeutic target for the clinical management of CRC. However, more mechanistic studies are required to further support this conclusion. Besides, as a secretable protein (Tsutsumi & Neckers, 2007), the serum HSP90 abundance of CRC patients could also be measured to evaluate the application in liquid biopsy.

Owing to the importance of HSP90 in the regulation of different cellular proteins, it has become an attractive therapeutic target for many cancers (Schopf, Biebl & Buchner, 2017). Currently, a number of specific HSP90 inhibitors have been evaluated in preclinical or clinical trials (Kryeziu et al., 2019). The first-generation inhibitors are mainly natural products, including Geldanamycin (GA) and radicicol (RD), which competitively bind with the N-terminal of ATP-binding pocket and prevent the ATP or ADP binding of HSP90 (Roe et al., 1999). However, clinical applications of the first-generation inhibitors were limited due to physicochemical properties and hepatotoxicity (Neckers & Workman, 2012). Therefore, more attentions were paid on the derivatives of GA and RD, so-called the second-generation inhibitors, including 17-AAG (tanespimycin), 17-DMAG (alvespimycin), IPI-504 (retaspimycin hydrochloride), KF25706 and KF58333 (Miyata, 2005; Porter et al., 2009). In spite of better solubility and lower toxicological properties, none of these inhibitors have been approved as a new drug due to the unsatisfactory antitumor effects (Soga, Akinaga & Shiotsu, 2013).

General reasons for the unsatisfactory therapeutic effects of HSP90 inhibitors have been described. One possible explanation is that the overexpression of some related proteins, such as Heat shock factor protein 1 (HSF1), P-glycoprotein 1 (P-gp) and UDP glucuronosyltransferase 1A (UGT1A), may be associated with resistance of HSP90 inhibitors (Kryeziu et al., 2019). However, the mechanistic aspects of HSP90 are still need to be investigated for improving the efficacy of HSP90 inhibitors.

Hence, the current study screened out 85 genes which were positively correlated with the expression of HSP90 and performed functional enrichment analysis to explore the biological pathways and processes correlated with HSP90. According to the GO analysis, the 85 genes mainly enriched in three biological processes embedding metabolic process, single organism process and response to stimuli. Additionally, these genes showed significant enrichment in molecular function of binding and catalytic activity. Besides, KEGG analysis enriched these genes in pathways of proteasome and base excision repair.

Consistent with our results, HSP90 is a “stress sensor” that facilitating in several cellular functions through regulating of folding and assembly of its client proteins. In recent years, it has also been reported that HSP90 chaperones participated in oncogene-driven metabolic rewiring, which is one of the hallmarks of cancer cells (Condelli et al., 2019). Obviously, HSP90 is closely cooperate with ubiquitin-proteasome machinery to control protein homeostasis (Makhnevych & Houry, 2012). Moreover, DMAG, a HSP90 inhibitor, was reported to interfere with base excision repair and ATM-mediated DNA repair (Koll et al., 2008). Taken together, the functional enrichment analyses of these HSP90 related genes may help us have a better understanding of HSP90 functions and implicate the application of rational drug combinations.

Conclusions

Collectively, we found HSP90 expression was upregulated in CRC tissues compared with normal tissues, and its high expression was associated with poor survival in CRC patients. Additionally, HSP90 expression was the independent risk factor that influencing the OS of CRC patients. Furthermore, bioinformatics analyses showed that the co-expression genes of HSP90 mainly involved in metabolic process, proteasome and base excision repair. These results suggested that high HSP90 expression may be used as an indicator of poor prognosis and predicted its roles in therapy for CRC patients.

Supplemental Information

Supplemental Information 1 Analysis of HSP90 genes expression levels from GEPIA with error bars.

Click here for additional data file.

Supplemental Information 2 A total of 85 genes screened from the TCGA dataset which were positively correlated with the expression of HSP90.

Pearson’s correlation coefficient ≥ 0.4, P < 0.0001.

Click here for additional data file.

Supplemental Information 3 The correlation coefficients between HSP90 expression and other mRNA expression.

Pearson’s correlation coefficient of all the genes from TCGA dataset.

Click here for additional data file.

Supplemental Information 4 The Pearson’s correlation coefficients of these 85 genes calculated by both GEPIA and TCGA datasets.

Click here for additional data file.

Supplemental Information 5 Raw data of patient clinicopathological features.

Click here for additional data file.

Supplemental Information 6 Immunohistochemical results of HSP90 expression in CRC and normal tissues.

Click here for additional data file.

We thank Dr. Lin Feng for Bioinformatic analysis of HSP90 related genes.

Additional Information and Declarations

Competing Interests

Author Contributions

Human Ethics

Data Availability

The authors declare that they have no competing interests.

Shuming Zhang performed the experiments, analyzed the data, prepared figures and/or tables, authored or reviewed drafts of the paper, approved the final draft.

Shichao Guo conceived and designed the experiments, authored or reviewed drafts of the paper, approved the final draft.

Zhangfu Li analyzed the data, contributed reagents/materials/analysis tools, authored or reviewed drafts of the paper, approved the final draft.

Dan Li conceived and designed the experiments, analyzed the data, prepared figures and/or tables, authored or reviewed drafts of the paper, approved the final draft.

Qimin Zhan conceived and designed the experiments, authored or reviewed drafts of the paper, approved the final draft.

The following information was supplied relating to ethical approvals (i.e., approving body and any reference numbers):

The ethics committee of National Cancer Center /Cancer Hospital of Chinese Academy of Medical Sciences and Peking Union Medical College granted Ethical approval to carry out the study (Ethical Application Ref: NCC2018-071).

The following information was supplied regarding data availability:

The immunohistochemistry assay raw data is available at Figshare: Guo, Shichao (2019): Raw data.xlsx. figshare. Dataset. DOI 10.6084/m9.figshare.9975689.v1

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
