# Peer review of "High expression of HSP90 is associated with poor prognosis in patients with colorectal cancer"

_PeerJ, doi:10.7717/peerj.7946_

## Round 0.1 · original submission · Major Revisions

The reviewers have a few concerns about the 85 gene. I would request the authors provide more details about the data filtering cutoff and follow-up functional analysis.

Reviewer 1 ·

Basic reporting

The manuscript is well written and easily understandable but still some corrections needed (ex line no 175). Good number of relevant literature references included. Article structure is fine. In the introduction, removing the second paragraph will not affect and will improve the flow of the manuscript. In the figure 1, I am suggesting to include error bars. Also it should be included that the number of samples/sequences considered in each cancer type to generate figure1

Experimental design

No comment

Validity of the findings

no comment

Additional comments

No comment

·

Basic reporting

In this manuscript, the authors confirmed that HSP90 is expressed in higher level in colorectal cancer (CRC) tissues than para-cancer normal tissues, and they demonstrated that high expression of HSP90 is associated with poor prognosis of CRC patients. They also looked at 85 genes correlated with HSP90 in expression, and did GO analysis and KEGG analysis. This study looks simple, clear and straightforward.

Experimental design

no comment

Validity of the findings

no comment

Additional comments

Major comments:
1. Line 160 the definition 'Pearson corr. >= 0.04, P<0.0001' vs. Line 223 '|Pearson corr. coef.| > 0.4' for positively correlated genes are very different. Which one is actually used? And if absolute value of Pearson corr. coef. is used, it should be only described as 'correlated' but not 'positively correlated'.
2. Line 176-177, 'Further separated analysis ... also consistent with above results (Fig 1C)'. Fig 1A-1C seems to be all generated by GEPIA, thus they are based on exactly the same data source. Therefore, 'Further ... also consistent' might be a little misleading and impropriate here. Do the highlighted (in red) cancer types in Fig 1B all have statistically significant differences as shown in Fig 1C? What statistical test is used for Fig 1B and 1C?
3. Line 188, 'Statistical analysis showed', what statistical test is used here? Since the score index is set by multiplication and its distribution looks deviated from normal distribution, I suppose the authors should use the nonparametric Mann-Whitney U test (i.e. Wilcoxon rank-sum test) here.

Minor comments:
1. Line 39, abbreviation 'CRC' is not explained in its first appearance.
2. Line 42, 'colorectal cancer tissues (CRC)' might be better as 'colorectal cancer (CRC) tissues'
3. Line 52-54, 'Moreover, 85 genes ... assigned to ... xxx GO terms' looks of little meaning or information.
4. Line 144, since 'GEPIA' is actually used to generate all figures of Fig 1, it may deserve a better citation.
5. Line 175, 'Fog Fig 1B' might be typo of 'Fig 1B'.
6. Line 182, 'score index' might be better as 'HSP90 expression score index' for clarity.
7. Line 202, 'M category' might be better explained.
8. Line 223, '85 genes were positively correlated with the expression of HSP90' might be better listed in a (supplementary) table. The GO terms listed in Fig 5 and 6 look so general and thus with little information.
9. Line 252, 'TNM stage' seems to be equivalent to 'AJCC stage'; 'TNM' is only used once here, while 'AJCC' is used many times elsewhere. And both abbreviations are not explained.
10. Figure 1, the abbreviations of cancer types are not explained.
11. Figure 2B, y-axis label '-5' seems meaningless.
12. Figure 5, does the color legend for Pvalue mean the original (uncorrected) p-value? Why not show in q-value, just like that in Fig 6B? And in Fig 6B, because the color legend has most range of insignificant q-values, it is not clear whether the top two enriched terms are actually still statistically significant or not after multiple-test correction.

·

Basic reporting

No comment

Experimental design

This research work is covered within the Aims and Scope of the journal.

The research questions are well defined, relevant & meaningful.

Rigorous investigations on the tissue biopsies were performed to a high technical & ethical standard.

Validity of the findings

Conclusions are well stated, linked to original research question & limited to supporting results.

Additional comments

This study designed to test the single gene HSP90 expression in CRC patients and healthy controls. Before performing this study, the authors tested in GEPIA databases, whether the HSP90 is the right target genes and identified that HSP90 is highly expressed in various cancers. The following questions need to be addressed.

1. The P-values for 10 cancer types for hsp90 is missing in the results.
2. Provide boxplot and correlation for HSP90 gene expression comparing the 99 CRC patients and 81 healthy controls. Are they significant between cases and controls?
3. The authors performed the chi-squre test and fisher’s exact test to see the significant differences between clinicopathological characteristics of CRC patients. What about simple t-test between the groups and the HSP90 expression.
4. The author selected 85 co-expressed genes from TCGA dataset and performed the pathway analysis. What about the co-expression of HSP90 in GEPIA? Does the author detect the same 85 genes that are co-expressed in HSP90 or those 85 genes are specific to TCGA?
5. How significantly those 85 genes are enriched in Figure 5B-5C, are they significant by p-value?
6. Figure legends were not provided

---

## Round 0.2 · accepted · Accept

We appreciate the revised manuscript. Please ensure all your figures meet the publication standard.

Reviewer 1 ·

Basic reporting

No comment

Experimental design

no comment

Validity of the findings

no comment

·

Basic reporting

The authors addressed all the questions raised by the reviewer

Experimental design

The authors addressed all the questions raised by the reviewer

Validity of the findings

The authors addressed all the questions raised by the reviewer

Additional comments

The authors addressed all the questions raised by the reviewer.